# UniMotion: A Unified Motion Framework for Simulation, Prediction and Planning

Nan Song[1]    Junzhe Jiang[1]    Jingyu Li[1,2]    Xiatian Zhu[3]    Li Zhang[1,2*]

[1]School of Data Science, Fudan University
[2]Shanghai Innovation Institute    [3]University of Surrey

https://github.com/LogosRoboticsGroup/UniMotion

## Abstract

Motion simulation, prediction and planning are foundational tasks in autonomous driving, each essential for modeling and reasoning about dynamic traffic scenarios. While often addressed in isolation due to their differing objectives, such as generating diverse motion states or estimating optimal trajectories, these tasks inherently depend on shared capabilities: understanding multi-agent interactions, modeling motion behaviors, and reasoning over temporal and spatial dynamics. Despite this underlying commonality, existing approaches typically adopt specialized model designs, which hinders cross-task generalization and system scalability. More critically, this separation overlooks the potential mutual benefits among tasks. Motivated by these observations, we propose **UniMotion**, a unified motion framework that captures shared structures across motion tasks while accommodating their individual requirements. Built on a decoder-only Transformer architecture, UniMotion employs dedicated interaction modes and tailored training strategies to simultaneously support these motion tasks. This unified design not only enables joint optimization and representation sharing but also allows for targeted fine-tuning to specialize in individual tasks when needed. Extensive experiments on the Waymo Open Motion Dataset demonstrate that joint training leads to robust generalization and effective task integration. With further fine-tuning, UniMotion achieves state-of-the-art performance across a range of motion tasks, establishing it as a versatile and scalable solution for autonomous driving.

## 1 Introduction

Motion understanding is a cornerstone of autonomous driving, supporting essential capabilities such as interactive behavior modeling, motion analysis, and decision-making for the ego vehicle. Core motion tasks include motion simulation, trajectory prediction, and ego planning. Among them, trajectory prediction [1, 2, 3] and ego planning [4] are critical intermediaries between perception modules and control systems in the driving pipeline, while motion simulation [5] generates rich and diverse agent behaviors, serving both development and evaluation for motion models. The complexity of real-world driving, characterized by uncertain behaviors and densely interactive environments, makes these tasks challenging.

Accordingly, recent research has produced a wide range of task-specific models to optimize performance for its designated objective. Trajectory prediction and planning models have primarily focused on comprehensive representation learning [6, 7, 8, 9], along with a growing emphasis on precise waypoint estimation [10, 11, 12, 13]. In contrast, motion simulation has leaned heavily on

---

[*]Li Zhang (lizhangfd@fudan.edu.cn) is the corresponding author.

39th Conference on Neural Information Processing Systems (NeurIPS 2025).

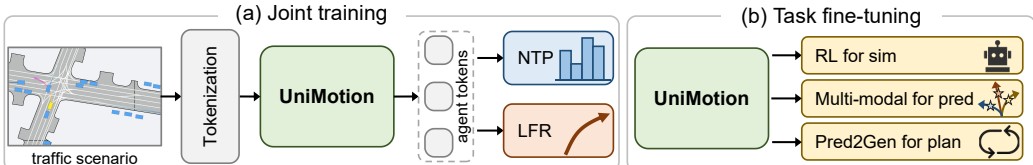

Figure 1: Overview of our UniMotion pipeline. (a) We train jointly our model with combined generative and forecasting supervision, resulting in a multi-task model that can simultaneously address all kinds of motion tasks in autonomous driving. (b) We further adopt dedicated fine-tuning strategies for each task, producing task-specific models to promote specialization.

generative modeling. Inspired by the success of Large Language Models (LLMs), recent GPT-style architectures [14, 15] have been applied to simulate agent behaviors. Furthermore, this paradigm has also been introduced into prediction [16, 17] and planning [13] tasks to achieve progressive and more accurate regression.

While these task-customized models have proven effective for different motion tasks, it is challenging to transfer models and learned knowledge across tasks. Considering that these tasks are essentially similar due to their reliance on interaction modeling and motion reasoning, our goal is to build a general model that encourages knowledge sharing among motion tasks and enables cross-task generalization. To achieve this aim, we revisit the formulation of motion tasks and propose a unified motion framework for autonomous driving. We begin by abstracting existing motion tasks into two fundamental categories: *diverse motion generation* and *long-range trajectory forecasting*, which serve as a conceptual foundation for unification. From this, we develop **UniMotion**, a unified framework based on a decoder-only Transformer, chosen for its simplicity, scalability, and strong generative capacity across tasks. By incorporating task-aware interaction modes and training strategies (see Figure 1(a)), UniMotion enables joint training across simulation, prediction and planning, fostering shared representations and efficient multi-task learning. To further enhance task proficiency, we also introduce dedicated fine-tuning strategies (see Figure 1(b)) that adapt the unified model to specific objectives, improving performance and practical deployment.

Our **contributions** are summarized as follows: **(i)** We abstract motion tasks into two core categories and propose UniMotion, a unified Transformer-based framework that jointly models simulation, prediction and planning to promote generality and cross-task knowledge sharing. **(ii)** We design effective fine-tuning strategies that specialize the jointly trained model for individual tasks, allowing the model to master task-specific expertise. **(iii)** Extensive experiments on the Waymo Open Motion Dataset (WOMD) show that UniMotion achieves competitive performance under joint training and sets new state-of-the-art results when fine-tuned for specific tasks.

## 2 Related work

### 2.1 Motion tasks in autonomous driving

Existing motion tasks in autonomous driving can be roughly categorized into three types according to their objectives: motion simulation, trajectory prediction, and ego planning. We provide a detailed discussion for each type in the following.

**Motion simulation** focuses on simulating diverse and complex motion patterns, augmenting training samples and offering off-board assessment for autonomous driving algorithms. To effectively measure simulation quality, Sim Agents [5] first proposes a benchmark with comprehensive metrics, expecting models to progressively evolve the motion states of all traffic agents. Motivated by the performance of LLMs, existing methods [14, 15] tokenize the agent trajectories and map elements, and employ a scalable decoder-only model to perform auto-regressive trajectory generation. Besides, CATK [18] utilizes closed-loop supervised finetuning with Closest Among Top-K rollouts, which mitigates covariate shift in open-loop behavior cloning and further improves these GPT-style methods. In contrast, UniMM [19] introduces a unified mixture model with both continuous mixture models and discrete GPT-style models to address simulation.

**Trajectory prediction** anticipates models to rapidly estimate the future trajectories of traffic participants, providing prior knowledge for subsequent decision-making. As a foundation task in autonomous driving, trajectory prediction has been widely explored [6, 7, 8, 11] with Transformer frameworks. Recent methods [20, 21, 22] uncover detailed temporal information and achieve more accurate prediction. Moreover, there are methods utilizing particular strategies for impressive performance improvements, such as model pre-training [23, 24] and trajectory post-refinement [10, 12].

**Ego planning** is responsible for giving final driving trajectories for ego vehicle according to the driving environment and upstream results. Although rule-based models [25, 26] still hold a crucial position, learning-based methods have emerged and exceeded. PlanTF [27] and PLUTO [28] effectively alleviate the limit of imitation-based planners through designed model architecture and training strategies. Besides, BeTopNet [9] further improves planning performance by explicitly representing behavioral topology among multi-agent future states.

## 2.2 GPT-style motion models

GPT-style LLMs have demonstrated strong capabilities in language tasks, which stimulates the emergence of similar architectures in autonomous driving [29, 30]. Given that both trajectories and text share a sequential structure, it is natural to introduce analogous modules and designs into motion models. In particular, the aforementioned simulation methods directly utilize decoder-only models with next-token prediction for simulation tasks, generating diverse multi-agent trajectory sets. There are also some prediction and planning methods adopting modified auto-regressive decoding. For example, MotionLM [17] and Trajeglish [16] introduce a causal decoder for iterative trajectory prediction, and similarly, CarPlanner [13] adopts consistent auto-regressive planning with reinforcement learning. In contrast, we unify all motion tasks under a GPT-style model, fully exploiting the capability of this framework.

# 3 Methodology

## 3.1 Preliminary

**Task definition.** Motion understanding in autonomous driving focuses on three tasks: simulation, prediction and planning. All these tasks share the same traffic scenario input, including historical agent states $\mathcal{A}_h$ and a high-definition (HD) map $\mathcal{M}$ of driving scenes. **Motion simulation** is typically designed to support other motion tasks. Given $\mathcal{A}_h$ as initial motion states, it aims to iteratively and concurrently simulate multi-agent states $\{\mathcal{A}_s^i\}_{i=1}^K$, where $K$ represents the number of generated rollouts. Simulation aims to exhibit high short-term generation diversity. In contrast, **trajectory prediction** provides distant future trajectories $\mathcal{A}_f$ of traffic agents over a span of time $T_f$. In real-world scenarios, we expect the predictors to generate the results directly and efficiently, rather than relying on segment-by-segment auto-regressive generation [16, 17]. Accordingly, this task evaluates the ability of models to perform accurate long-range predictions. **Ego planning** aims to determine feasible trajectories for the ego vehicle by referencing surrounding agents. It can be viewed as first predicting the behaviors of other traffic participants, followed by the progressive generation of the ego vehicle's trajectory. In light of the above observations, we summarize motion tasks into two fundamental components: *diverse motion generation* and *long-range trajectory forecasting*. As motivated earlier, we strive to integrate these tasks into a unified framework and jointly develop both capabilities in a single model, thereby enhancing overall motion understanding and fostering mutual complementarity between tasks.

**Input representation.** To ensure the generalization of representations across tasks, we standardize the input format via tokenization following [14, 15]. Concretely, the trajectories of all agents are segmented at fixed time intervals and normalized to form a trajectory set, which is then clustered to construct the agent token vocabulary $\mathcal{S}_a$. Similarly, map tokens $\mathcal{S}_m$ are generated by dividing original map polylines into fixed-length segments followed by clustering. The corresponding agent token embeddings $E_{sa} \in \mathbb{R}^{K_a \times C}$ and map token embeddings $E_{sm} \in \mathbb{R}^{K_m \times C}$ obtained by applying MLP modules to $\mathcal{S}_a$ and $\mathcal{S}_m$, respectively, where $K_{(\cdot)}$ and $C$ are the number of tokens and the dim of feature embeddings. Subsequently, the agent states $\mathcal{A}$ and a HD map $\mathcal{M}$ are tokenized based on geometric similarity of segments, forming the input agent embeddings $E_a \in \mathbb{R}^{N_a \times T \times C}$ and map

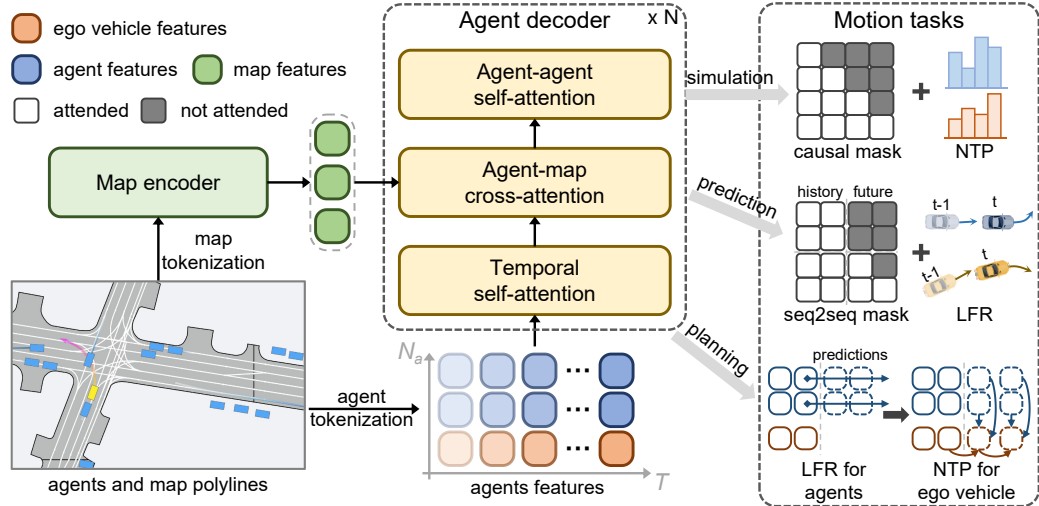

Figure 2: Overview of UniMotion architecture. It takes as input tokenized agent trajectories and map polylines, and adopts a decoder-only structure equipped with an additional map encoder to yield motion states. By employing task-specific attention masks and training objectives, UniMotion is empowered to flexibly and effectively address multiple motion tasks simultaneously.

embeddings $E_m \in \mathbb{R}^{N_m \times C}$, where $N_a$ and $N_m$ denote the number of agents and map segments, and $T$ denotes the time frames.

## 3.2 UniMotion for multiple motion tasks

As depicted in Figure 2, the overall design of our **UniMotion** follows a decoder-only architecture for agent trajectory generation and prediction, with an additional map context encoder. For each specific motion task, we introduce dedicated interactive patterns and training objectives for specialization, but enable weight sharing to learn general representations.

### 3.2.1 Scene context encoding and motion decoding

To ensure the robustness and generalizability of the model, we employ the Transformer architecture with relative positional embedding following [8, 14, 15], which remains invariant under coordinate transformations. Specifically, we first stack several self-attention modules to encode map embeddings, yielding map features $F_m$. Then, agent embeddings $E_a$, along with $F_m$, are further delivered to a agent Transformer decoder constructed with factorized attentions, which are composed of temporal self-attention, agent-map cross-attention and agent-agent self-attention. This can be formulated as:

$$
\begin{aligned}
F_a &= \text{MHSA}_t(\text{Q} = F_a, \text{K} = \text{V} = F_a + \mathcal{R}_t(P_a)), \\
F_a &= \text{MHCA}_{a-m}(\text{Q} = F_a, \text{K} = \text{V} = F_m + \mathcal{R}_{a-m}(P_a, P_m)), \\
F_a &= \text{MHSA}_{a-a}(\text{Q} = F_a, \text{K} = \text{V} = F_a + \mathcal{R}_{a-a}(P_a)),
\end{aligned}
\tag{1}
$$

where $\mathcal{R}$ is the relative positional embedding for spatial and temporal position information $P_{(\cdot)}$, and the initial $F_a$ is directly derived from $E_a$. This decoder module enables the thorough interaction among agents and between agents and road elements, finally giving step-wise motion states.

### 3.2.2 Joint training across motion tasks

Following the designs of multi-task language models, we apply customized interactive attention masks for different motion tasks and introduce tailored training objectives to improve both step-wise generation and long-range prediction. At inference time, we adaptively leverage the model's capabilities according to the specific task.

**Motion simulation.** The motion simulation task closely resembles language generation, which aims to iteratively select suitable tokens from motion vocabulary. Accordingly, we adopt a causal attention mask and employ Next-Token Prediction (NTP) as the learning strategy, modeling the generation of

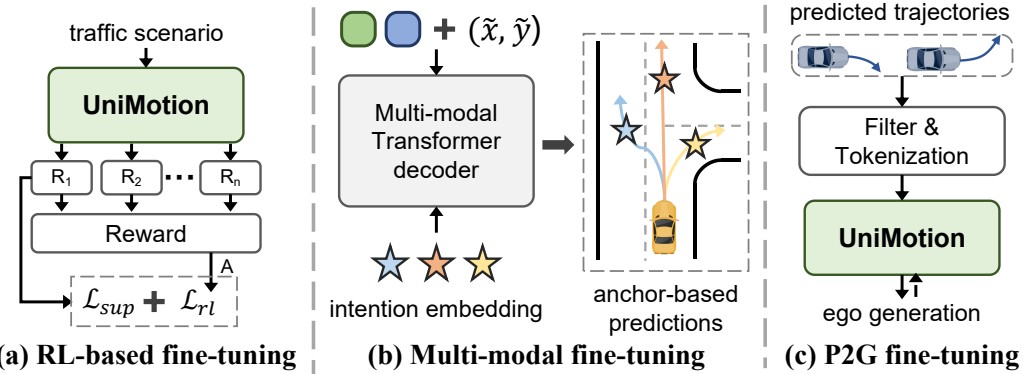

Figure 3: Illustration of task-specific fine-tuning strategies. We adopt (a) **RL-based fine-tuning** to improve simulation likelihood while maintaining closed-loop consistency. (b) **Multi-modal fine-tuning** is introduced to refine the multi-modal behavior of predictions. With reference to the planning inference, we utilize (c) **Pred2Gen fine-tuning** to address distribution mismatch.

the next agent token $\mathcal{A}_t$ at step $t$ from the conditional probability $P(\mathcal{A}_t \mid \mathcal{A}_{<t})$. During inference, motion processes for all agents are simultaneously simulated through auto-regressive generation.

**Trajectory prediction.**    We regard trajectory prediction as a sequence-to-sequence task, which maps historical segments to future trajectories. Tokens within the historical segments are allowed to attend only to each other, while future tokens can attend both to preceding tokens within the future segment and to all tokens in the historical segments. Moreover, we propose Long-range Future Regression (LFR) to meet the strict requirements of prediction task on accuracy. Specifically, we replace token classification with normalized trajectory regression, and it predicts a complete long-range future trajectory instead of short segments with token-level length, in which case future trajectories $\mathcal{A}_f$ can be derived from:

$$\mathcal{A}_f = \mathrm{MLP}(F_a), \ \mathcal{A}_f \in \mathbb{R}^{N_a \times T \times T_f \times 2}. \tag{2}$$

Notably, all these tokens are utilized for dense supervision in the training phase, while we only need to perform single-pass reasoning at current step for inference.

**Ego planning.**    As mentioned in Section 3.1, we consider planning task to be the combination of generation and prediction. Hence, the training objectives for simulation and prediction naturally benefit the planning task, without requiring any additional training strategies. After training, we adopt a two-stage inference approach for ego trajectory generation. Concretely, we first jointly predict future trajectories for all traffic participants, and then tokenize the future parts of these predictions to form the future tokens. Given the predicted future tokens of surrounding agents, we then progressively generate the planning trajectory of ego vehicle.

### 3.2.3    Task-specific fine-tuning strategies

Despite the capability of solving multiple tasks simultaneously, models are typically expected to exhibit higher specialization for individual tasks in real-world applications. To this end, we introduce task-specific fine-tuning strategies to further enhance the specialization of UniMotion. Overall, we prefer not to introduce additional parameters during fine-tuning stage. According to this principle, we utilize reinforcement learning-based fine-tuning for simulation and two-stage pred2gen finetuning for planning. Additionally, trajectory prediction requires to produce multi-modal outputs, which will significantly increase the computational burden under dense supervision. To alleviate this, we employ a multi-modal decoder to support the prediction process as a compromise.

In line with the purpose of Reinforcement Learning from Human Feedback (RLHF) in LLMs, we enhance the model to generate trajectories that align with human driving behavior and traffic rules, which cannot be constrained through direct supervision alone. Hence, we introduce RL-based fine-tuning for simulation. As shown in Figure 3(a), we first generate $n$ rollouts of driving scenes as a group following GRPO [31], with activating the gradient for only one rollout. Subsequently,

rewards $R$ are computed using the same algorithms as the metrics in Sim Agents [5], which consist of kinematic reward and collision reward for simplicity. We formulate this as:

$$R = \exp(\frac{1}{T}\sum_{i=1}^{T}\log q^{\text{gt}}) + \mathbf{1}\{\text{not collision}\}, \tag{3}$$

where the kinematic reward is defined as the exponentiated log-likelihood of ground truth samples under the distribution induced by corresponding generated trajectories, whereas the collision reward is an indicator function. Then, the advantage $A$ is derived from the normalized rewards within the group of each agent. Notably, we only supervise one rollout to reduce computational overhead. More details are provided in Section 3.3 and supplementary materials.

To improve multi-modality of predicted trajectories, we introduce a lightweight transformer decoder similar to [11, 32] as illustrated in Figure 3(b). At this stage, we only focus on the agents of interest that are utilized for evaluation, and perform prediction within perspective local system. To this end, the coordinate-invariant features are transformed into the local agent features $F_a^*$ and map features $F_m^*$ through adding the local normalized position content encoded by MLP layer. For each interested agent, the estimated multi-modal trajectories $a_f^* \in \mathbb{R}^{N_{mo}\times T_f\times 2}$ for $N_{mo}$ modes can be derived by:

$$a_f^* = \text{Transformer}(\text{Q} = F_{mo}, \text{K} = \text{V} = \{F_a^*, F_m^*\}), \tag{4}$$

where $F_{mo}$ denotes the mode features encoded from intention points, and $F_a^*$ and $F_m^*$ are with respect to current interested agent. Benefiting from distinct intention points, our model can provide more diverse trajectory choices after fine-tuning.

During planning inference, the predicted trajectory tokens for surrounding agents will participate in and significantly affect ego generation, inconsistent with the independent training process. To alleviate potential distribution mismatch, we further fine-tune ego generation using the predicted results. As shown in Figure 3(c), the incorrectly predicted trajectories are first filtered out and replaced by the ground truth. Then, we tokenize all predictions and feed them into our network for ego generation, which is fine-tuned with end-to-end supervision.

## 3.3 Model training

During the joint training process, we mainly adopt NTP classification loss $\mathcal{L}_{\text{ntp}}$ between next tokens $\mathcal{A}_s$ and corresponding ground truth $\hat{\mathcal{A}}_s$ for generation supervision, and LFR regression loss $\mathcal{L}_{\text{lfr}}$ between $\mathcal{A}_f$ and ground truth $\hat{\mathcal{A}}_f$ for trajectory prediction supervision. We combine these two individual losses with equal weights to form the overall loss $\mathcal{L}$, formulated as follows:

$$\mathcal{L} = \mathcal{L}_{\text{ntp}}(\mathcal{A}_s, \hat{\mathcal{A}}_s) + \mathcal{L}_{\text{lfr}}(\mathcal{A}_f, \hat{\mathcal{A}}_f), \tag{5}$$

where we employ the cross-entropy loss and the smooth-L1 loss for $\mathcal{L}_{\text{ntp}}$ and $\mathcal{L}_{\text{lfr}}$, respectively.

During the fine-tuning stage, we employ different supervision strategies for each motion task. For simulation fine-tuning, we follow [33] to simplify the GRPO but still preserve the consistency constraint through closed-loop supervision on the gradient-activated rollout $\mathcal{A}_r \in \mathbb{R}^{N_a\times T}$. Besides, in light of efficiency concerns, we only use $\mathcal{A}_r$ rather than all rollouts to perform a single policy update per iteration. The simulation fine-tuning loss is:

$$\mathcal{L}_{\text{ft-sim}} = \mathcal{L}_{\text{ce}}(\mathcal{A}_r, \hat{\mathcal{A}}_r) + w_s\mathcal{L}_{\text{p}}(\mathcal{A}_r, A), \tag{6}$$

where $\mathcal{L}_{\text{ce}}$ is the cross-entropy loss between $\mathcal{A}_r$ and ground truth $\hat{\mathcal{A}}_r$, $\mathcal{L}_{\text{p}}$ is the policy utilized in GRPO with taking $\mathcal{A}_r$ and advantage $A$ as inputs, and $w_s$ is a balancing factor. The prediction loss consists of Gaussian NLL loss $\mathcal{L}_{\text{nll}}$ to supervise multi-modal trajectories $a_f^*$ and cross-entropy loss for mode classification. Meanwhile, the model also learns to regress the local future trajectories $\mathcal{A}_f^*$ for other agents as auxiliary supervision. The loss can be formulated as:

$$\mathcal{L}_{\text{ft-pred}} = w_{\text{pr}}\mathcal{L}_{\text{lfr}}(\mathcal{A}_f^*, \hat{\mathcal{A}}_f^*) + \mathcal{L}_{\text{nll}}(a_f^*, \hat{a}_f^*) + \mathcal{L}_{\text{ce}}. \tag{7}$$

In addition, we simply supervise the generation $a_{\text{ego}}$ for ego vehicle and the predictions $\mathcal{A}_f^{\text{surr}}$ for surrounding agents in the planning fine-tuning stage as:

$$\mathcal{L}_{\text{ft-plan}} = w_{\text{pl}}\mathcal{L}_{\text{lfr}}(\mathcal{A}_f^{\text{surr}}, \hat{\mathcal{A}}_f^{\text{surr}}) + \mathcal{L}_{\text{ntp}}(a_{\text{ego}}, \hat{a}_{\text{ego}}). \tag{8}$$

Table 1: Performance comparison on 2025 Waymo Open Sim Agents Challenge.

| Method | Realism Meta Metric ↑ | Kinematic Metrics ↑ | Interactive Metrics ↑ | Map-based Metrics ↑ | minADE ↓ |
|---|---|---|---|---|---|
| UniTAM | 0.7698 | 0.4869 | 0.7876 | 0.9085 | 1.3940 |
| UniTFormer | 0.7776 | 0.4892 | 0.7997 | 0.9140 | 1.3592 |
| LLM2AD | 0.7779 | 0.4846 | 0.8048 | 0.9109 | **1.2827** |
| UniMM [34] | 0.7829 | 0.4914 | 0.8089 | 0.9161 | 1.2949 |
| CATK [18] | 0.7846 | 0.4931 | **0.8106** | 0.9177 | 1.3065 |
| **UniMotion** | **0.7851** | **0.4943** | 0.8105 | **0.9187** | 1.3036 |

Table 2: Performance comparison on WOMD Prediction Leaderboard. For each metric, the best and second best results are highlighted in **bold** and underline, respectively.

| Method | minADE ↓ | minFDE ↓ | MR ↓ | mAP ↑ | Soft mAP ↑ |
|---|---|---|---|---|---|
| HDGT [35] | 0.5933 | 1.2055 | 0.1854 | 0.3577 | 0.3709 |
| MTR [11] | 0.6050 | 1.2207 | 0.1351 | 0.4129 | 0.4216 |
| MTR++ [36] | 0.5906 | 1.1939 | 0.1298 | 0.4329 | 0.4414 |
| MGTR [37] | 0.5918 | 1.2135 | 0.1298 | 0.4505 | 0.4599 |
| EDA [32] | **0.5718** | 1.1702 | 0.1169 | 0.4487 | 0.4596 |
| ControlMTR [38] | 0.5897 | 1.1916 | 0.1282 | 0.4414 | 0.4572 |
| RMP-YOLO [39] | 0.5737 | 1.1697 | **0.1160** | 0.4523 | **0.4673** |
| **UniMotion** | **0.5718** | **1.1643** | 0.1162 | **0.4534** | 0.4642 |

# 4 Experiments

## 4.1 Experimental settings

**Datasets and metrics.** To evaluate the performance of our method, we conduct extensive experiments on the Waymo Open Motion Dataset, comprising 486,995 training scenarios, 44,097 validation scenarios and 44,920 testing scenarios. Each scenario has an observation window of 9.1 seconds, consisting of 91 frames sampled at 10 Hz. Given 11-frame historical agent states and a HD map, the motion models are tasked with generating or predicting the future states of 80 frames. Additionally, prediction and planning tasks only perform one-step inference at the current time, while simulation task requires 32 simulations per scenario.

We employ respective benchmark metrics to assess our model. For simulation, the core metric is the Realism Meta Metric, which is utilized to measure the similarity between the simulations and the real-world distribution. It is calculated by the weighted sum of Kinematic Metrics for motion features, Interactive Metrics to capture the interaction among agents and Map-based Metrics to test the agent behavior on the map. For prediction, 6 predicted trajectories for each interested agent are expected for evaluation, with metrics including minADE (minimum Average Displacement Error), minFDE (minimum Final Displacement Error), Miss Rate, mAP and Soft MAP. In addition, due to the absence of an official benchmark for planning, we adopt the metrics proposed by [40] as measures.

**Implementation details.** We tokenize the trajectories and map elements following [15, 18], with 2,048 tokens for agents and 1,024 tokens for map. During the training phase, We train our models using the AdamW [41] optimizer with a total batch size of 48 on 8 NVIDIA A6000 GPUs for 30 epochs, with the weight decay set to 0.01. We initialize the learning rate of $5 \times 10^{-4}$, which is then decayed following a cosine annealing schedule. Regarding the fine-tuning stage, we employ the same learning rate for prediction, while a lower learning rate of $5 \times 10^{-5}$ for simulation and planning. Beside, we also fine-tune these tasks with the same epochs.

Each agent token covers 0.5 seconds, and each map token covers about 0.5 meters. After tokenization, the original 91-step trajectories in Waymo are compressed into 18 frames. For the training phase, the training time for full data is approximately 3.5 days, while using 20% of the data takes around 1 day in our experimental settings. In addition, we set the balance factor $w_s$, $w_{pr}$ and $w_{pl}$ to 0.1, 0.1 and 0.5, respectively.

Table 3: Performance comparison for Waymo planning.

| Method | Collision ↓ | Red light ↓ | Off route ↓ | Planning error ↓ @1s | @3s | @5s |
|---|---|---|---|---|---|---|
| IL | 5.469 | 2.772 | 4.816 | 0.175 | 1.416 | 4.194 |
| IL+Prediction | 3.930 | 1.670 | 4.542 | 0.127 | 0.892 | 2.901 |
| Sep. Plan+Pred. | 1.813 | 1.327 | 0.527 | 0.238 | 1.251 | 3.466 |
| DIPP [40] | 1.802 | **1.235** | 0.506 | 0.227 | 1.187 | 3.335 |
| **UniMotion** | **1.565** | 1.309 | **0.477** | **0.083** | **0.591** | **2.246** |

## 4.2  Comparison with state of the art

We compare the performance of our UniMotion with top-ranked competitors on Sim Agents [5], WOMD Prediction and Waymo Planning [40] benchmarks. The leaderboards for Sim Agents and WOMD Prediction are presented in Table 1 and Table 2, respectively. For simulation, our method achieve the best score of 78.51 on Realism Meta Metric. Benefiting from the kinematic reward of fine-tuning, the model is encouraged to generate trajectories with higher motion similarity to real-world situations, driving better Kinematic Metrics than other methods. However, the minADE metric exhibits relatively poorer performance, which can be attributed to the tokenization-based design's stronger emphasis on generation diversity at the cost of slightly reduced prediction accuracy. The leaderboard of the prediction task demonstrate that our method achieves promising results across almost all metrics, except for Soft mAP. Regarding the planning task, UniMotion has far outperformed most of previous approaches as depicted in Table 3, showing that our method stands distinctly ahead of others in terms of planning error. Moreover, to ensure generality and consistency, we supervise model training solely using logged trajectories. This mechanism leads to suboptimal performance in reacting to red lights, which can be mitigated by incorporating traffic rule constraints, as in [40].

## 4.3  General ablation study

We conduct ablation studies on the WOMD validation split to examine the effectiveness of each component in UniMotion. For efficiency, we use only 20% of the training data for ablation studies, while adopting the default experimental settings following Section 4.1 for all other aspects. We focus on the core modules here, while leaving task-specific ablation to the supplementary materials.

**Effects of NTP and LFR.**  As shown in Table 4, we simply assess the effectiveness of NTP and LFR in our network on simulation and prediction tasks. The first and second rows demonstrate the results of training UniMotion using only NTP or LFR, respectively. Despite the absence of dedicated supervision for another one of tasks, we report all relevant metrics to compare with joint training and showcase the model's ability to handle multiple tasks. It can be observed that the model benefits from both NTP and LFR in generation and prediction, while it performs poorly on another task. In the third row, we implement a joint training strategy with these two types of supervision, simultaneously endowing the model with diverse generative and long-range predictive capabilities. Besides, we find that joint training further improves prediction accuracy compared to using LFR alone, which we attribute to the diverse generation directions serving as targets to guide the prediction process.

Table 4: Ablation on joint training with NTP and LFR. "Kin."/"Inter.": Kinematic/Interactive metrics.

| NTP | LFR | Simulation | | | | Prediction | | |
|---|---|---|---|---|---|---|---|---|
| | | Kin. | Inter. | Map | minADE | minADE | minFDE | mAP |
| ✓ | | 0.4884 | 0.7961 | **0.9148** | 1.4074 | 0.7697 | 1.5547 | 0.2629 |
| | ✓ | 0.4401 | 0.7742 | 0.8949 | 1.2965 | 0.6668 | 1.3613 | 0.2935 |
| ✓ | ✓ | **0.4892** | **0.7968** | 0.9144 | **1.4011** | **0.6508** | **1.3296** | **0.3147** |

**Effects of task-specific fine-tuning.**  After training a preliminary model with a joint strategy, we further tailor it with designs specific to the target task. As demonstrated in Table 5, fine-tuning methods can intuitively bring consistent improvements across all motion tasks. For simulation, all metrics exhibit varying degrees of gains after fine-tuning, especially in the reward items. The 3rd and

4th rows of prediction task show that the mAP metric has been remarkably enhanced by introducing an extra anchor-based decoder, which can cover more potential future trajectories. In addition, noticeable improvements are also observed on the minADE and MR metrics, achieving the expected goals of fine-tuning. Regrading the planning task, the 5th row shows limited performance due to the distribution mismatch between predicted tokens and original trajectory tokens. After fine-tuning, the model is adapted to the two-stage process, and there are considerable improvements for all metrics.

Table 5: Ablation on task-specific fine-tuning for simulation, prediction and planning. Notably, we selected a set of metrics with substantial variation to facilitate comparison. Additionally, we fine-tune planning model with the data following [40] instead of 20% data.

| Task | ft. | Metrics | | | |
|---|---|---|---|---|---|
| | | Kinematic ↑ | Interactive ↑ | Map-based ↑ | minADE ↓ |
| Simulation | ✗ | 0.4892 | 0.7968 | 0.9144 | 1.4011 |
| | ✓ | **0.4939** | **0.8032** | **0.9159** | **1.3904** |
| | | minADE ↓ | minFDE ↓ | MR ↓ | mAP ↑ |
| Prediction | ✗ | 0.6508 | **1.3296** | 0.1626 | 0.3147 |
| | ✓ | **0.6413** | 1.3368 | **0.1493** | **0.3856** |
| | | Collision ↓ | Err @1s ↓ | Err @3s ↓ | Err @5s ↓ |
| Planning | ✗ | 1.7347 | 0.1326 | 0.7943 | 2.7809 |
| | ✓ | **1.5650** | **0.0834** | **0.5908** | **2.2455** |

## 4.4 Task-specific ablation study

In this section, we adopt the same experimental settings and conduct task-specific ablation studies on the simulation and prediction tasks to investigate the effects of different module designs.

**Effects of RL-based fine-tuning for simulation.** We present the effects of different constraints in Table 6. As shown in the 2nd and 3rd rows, closed-loop supervised fine-tuning leads to notable improvements, while relying solely on RL-based supervision, like DAPO [33], fails to produce satisfactory results. We attribute this to two primary factors: insufficient reward signals and limited model capacity.

Table 6: Ablation on RL-based fine-tuning . "con.": consistency constraint. "Kin."/"Inter.": Kinematic/Interactive metrics.

| con. | rl | Kin. | Inter. | Map | minADE |
|---|---|---|---|---|---|
| | | 0.4892 | 0.7968 | 0.9144 | 1.4011 |
| ✓ | | 0.4921 | 0.8019 | 0.9156 | 1.3933 |
| | ✓ | 0.4913 | 0.7974 | 0.9120 | 1.3967 |
| ✓ | ✓ | **0.4939** | **0.8032** | **0.9159** | **1.3904** |

**Effects of fine-tuning for prediction.** We present the effects of different fine-tuning strategies in Table 7. The comparison between the 2nd and 3rd rows demonstrates that although the end-to-end fine-tuning strategy (without intention anchors) can significantly reduces prediction errors, it severely compromises prediction diversity and confidence, leading to degraded mAP performance. Hence, we retain the intention anchors as part of the fine-tuning process.

Table 7: Ablation on prediction fine-tuning.

| Strategy | minADE | minFDE | MR | mAP |
|---|---|---|---|---|
| w/o ft. | 0.6508 | 1.3296 | 0.1626 | 0.3147 |
| ft. w/o intention | **0.6174** | **1.2639** | **0.1313** | 0.3389 |
| ft. w/ intention | 0.6413 | 1.3368 | 0.1493 | **0.3856** |

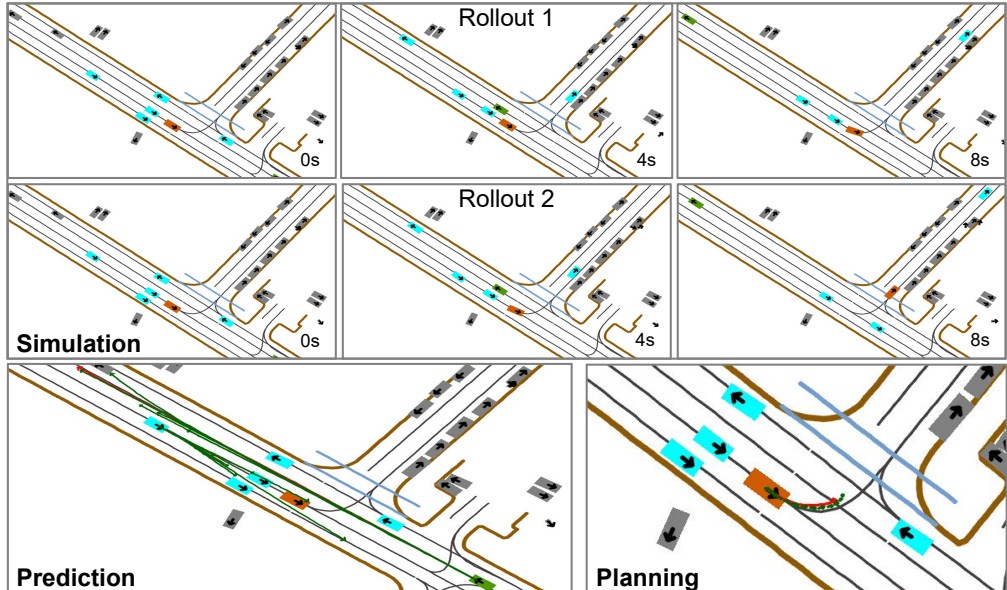

Figure 4: Qualitative results on WOMD validation split. The top and bottom rows present two rollouts of simulation and the results of prediction and planning, respectively. The red and green arrowed curves are ground truth and predicted trajectories.

## 4.5   Qualitative results

In Figure 4, we present qualitative results of our network on three motion tasks. The first two panels illustrate the diverse simulation results of all traffic agents at 0s, 4s, and 8s, respectively. The bottom-left figure depicts the multi-modal predictions for interested agents,while the bottom-right figure presents the ego trajectory generation process in a token-by-token manner. These qualitative results demonstrate our network's capability in handling multiple motion tasks.

## 5   Conclusion

In this work, we abstract mainstream motion tasks into two core types and aim to address all motion tasks within a unified framework, thereby promoting knowledge sharing and enhancing model generalization. Achieving this, we present UniMotion, an integrated model designed particularly to support joint training for these tasks. The critical components of our framework are distinct interactive modes and training strategies. Besides, we further perform task-specific fine-tuning for each task, fulfilling the requirement for expert-level models in real-world applications. Our extensive experiments on several task benchmarks comprehensively demonstrate that UniMotion is capable of unifying motion tasks, and the fine-tuned versions surpass the current state-of-the-art performance in the respective tasks.

**Limitations and future work.**   Although we employ a decode-only framework to ensure flexibility, the coupled spatial and temporal interaction and relative position embedding rely on detailed positional information of traffic elements, constraining the model from leveraging more LLM-related technologies. Moreover, after applying tokenization and conducting joint multi-task training, it is worthwhile to explore cross-dataset learning, which could lead to substantial improvements in joint training performance. Limited by computational resources and training time, we have not explored this technique in the present work.

## Acknowledgments

This work was supported in part by National Natural Science Foundation of China (Grant No. 62376060).

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

# A Details for simulation fine-tuning

After obtaining the generated rollout $\mathcal{A}_r$ and advantage $A$, we update policy gradient using simplified objective to ensure stable training of lightweight models, the gradient of which for token $t$ can be estimated as:

$$\nabla_\theta \log \mathcal{F}_\theta(\mathcal{A}_r[t]) \cdot A_{(\mathcal{A}_r)}[t], \tag{9}$$

where $\mathcal{F}_\theta$ denotes our trainable models. Besides, considering the relatively uniform rewards and the large number of iterations, we perform policy update for only one generation per iteration to promote efficient training.

# B Discussions

**Why we employ decoder-only Transformers as the backbone?** We opt for a decoder-only Transformer structure because it is better suited for multi-step generative tasks like simulation, where input and output lengths can vary. Conversely, the encoder-decoder structure, while excellent for sequence-to-sequence prediction tasks, cannot generalize as well across the diverse demands of our multi-task framework. Adopting a decoder-only architecture during joint training ensures stronger generalization capabilities across all our tasks.

**Why the joint training in UniMotion works well?** As detailed in Section 3.1, our primary objectives are NTP and LFR. Intuitively, this joint training paradigm fosters the learning of shared, rich intermediate representations across multiple tasks. This enables tokens within the model to attend simultaneously to both their immediate surroundings and more distant scene contexts, leading to a more holistic understanding of the environment.

Further, despite their apparent differences, the two task objectives possess crucial commonalities. Specifically, while LFR provides direct supervision on long-range logged trajectories, the initial motion direction embedded within these trajectories concurrently serves as the maximization objective for NTP. This underlying consistency and synergistic relationship between LFR and NTP are fundamental to ensuring stable joint training and preventing mode collapse, allowing the model to learn effectively from both detailed local cues and broad long-range dependencies.

# C More qualitative results

We provide more qualitative results of our framework in Figure 5 and Figure 6 on the validation set of Waymo Open Motion Dataset [1, 5].

# D Failure cases

Although our UniMotion achieves strong performance across multiple motion tasks, it still encounters certain failure cases. We provide qualitative results and analysis to offer a more comprehensive understanding, which we hope will support future efforts toward developing more robust and capable algorithms, as shown in Figure 7 and Figure 8.

In simulation task, there is still a possibility of collisions in complex scenario as illustrated in Figure 7 (a). Besides, we have also observed some unrealistic simulations in intersection scenarios as shown in Figure 7 (b), where agents remain stationary despite the green light. We provide the failure cases of prediction task in Figure 8 (a), which demonstrate that it is challenging to predict complex motion patterns, especially sudden turns. Similar issues are also observed in planning, particularly when dealing with sudden U-turns. Additionally, the model might yield reasonable trajectories that fail to reflect the driver's subjective intention.

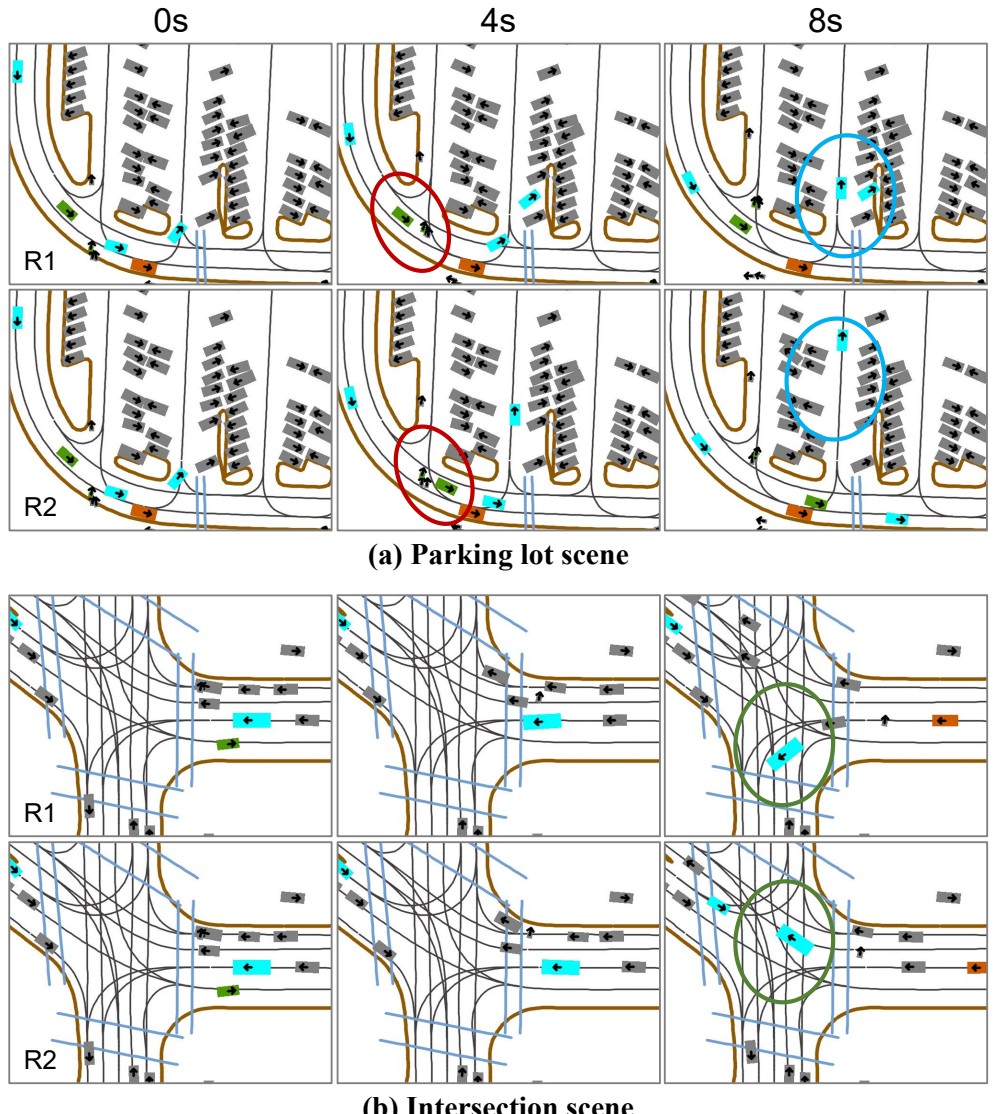

Figure 5: Qualitative results on the Sim Agents. We present several complex scenarios in autonomous driving, each accompanied by two rollouts. The circled areas highlight the key differences between the two rollouts.

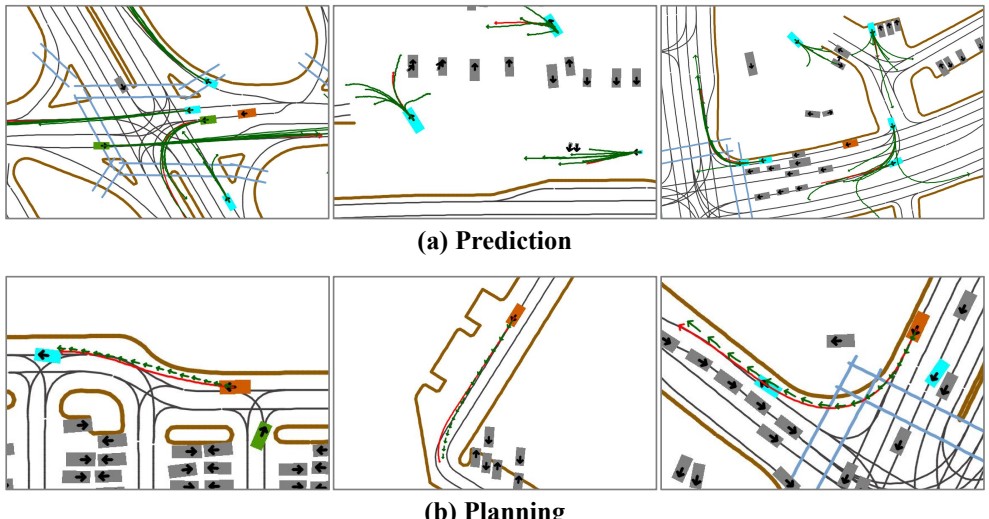

**(a) Prediction**

**(b) Planning**

Figure 6: Qualitative results on the WOMD for prediction and planning task.

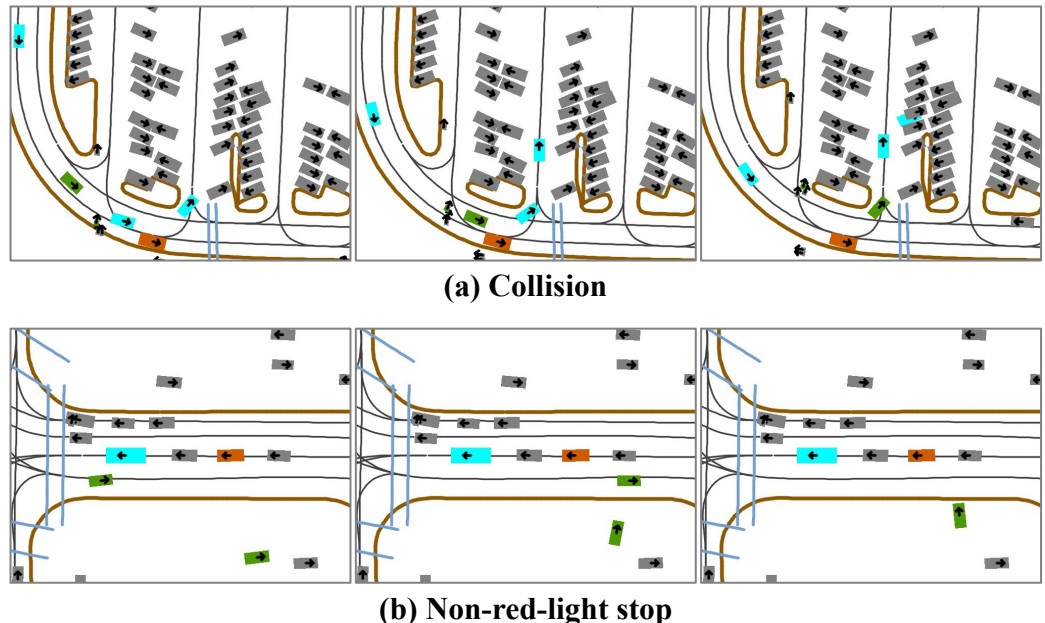

**(a) Collision**

**(b) Non-red-light stop**

Figure 7: Failure cases in simulation. In the first row, the model causes a collision in a crowded parking lot. In the second row, it incorrectly incorrectly stop without a red light at an intersection.

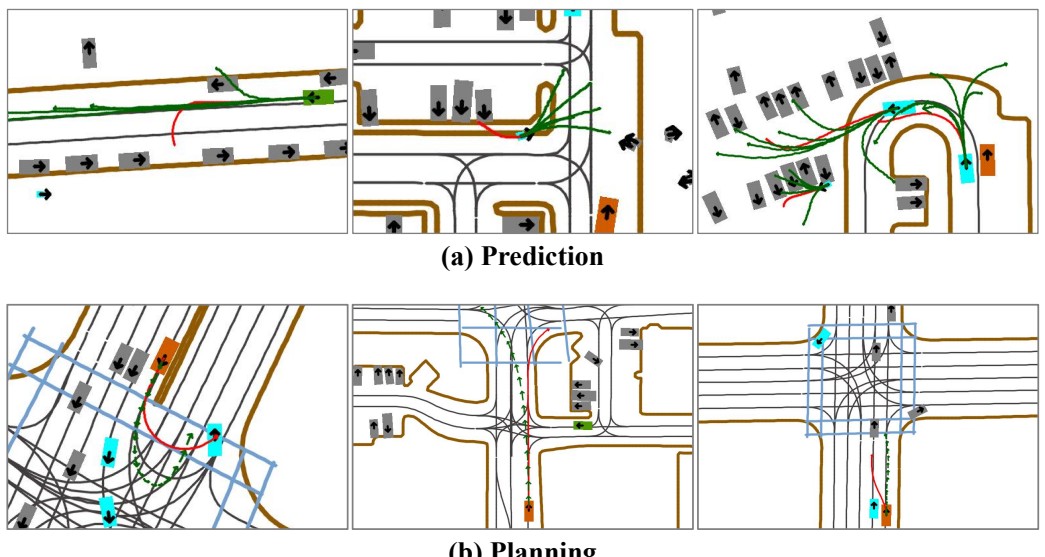

**(a) Prediction**

**(b) Planning**

Figure 8: Failure cases in prediction and planning. The model fails to accurately predict the trajectories of vehicles and pedestrians. In terms of planning, it struggles with scenarios where the targets involve drivers' subjective intentions.

