# OpenReview forum: "UniMotion: A Unified Motion Framework for Simulation, Prediction and Planning"
_NeurIPS.cc/2025/Conference — NeurIPS 2025 poster_

### Official Review · Reviewer_uSmG · 2025-06-07

**Clarity:** 3
**Significance:** 2
**Originality:** 2
**Rating:** 4
**Confidence:** 4

**Summary:**

This work allows a transformer-based model to unify the skills needed for three common self-driving tasks, including trajectory prediction, agent simulation, and ego-car planning. This base model then follows three distinct post-training/fine-tuning methods to tailor itself for the three tasks individually. The SOTA benchmark results on Waymo prediction, Sim Agent Challenge, and Waymo planning show the effectiveness of the recipe. Ablation study further highlights the effectiveness of each training design and the three post-training strategies.

**Questions:**

What is the size of the model? If it is too large, then we would still stick to individual solutions for specific tasks.

**Ethical Concerns:**

["NO or VERY MINOR ethics concerns only"]

**Final Justification:**

With the new results provided, the paper's claim is well-supported. I tend to accept this paper.

**Quality:**

2

**Strengths And Weaknesses:**

### Strength
1. The nature of the three common self-driving tasks is the same: trajectory prediction. Thus, they can be addressed together, which is a good motivation for this work.
2. The method itself follows the standard LLM training paradigm with transformer backbone, next-token prediction, and RL-based post-training. It makes it easy to reuse the mature infrastructure of LLM development for training a UniMotion model.
3. The experiment is thorough and shows SOTA performance, and the ablation study clearly shows why a certain module/process works.

### Weakness
1. The main weakness is the evaluation of planning ability. It is tested on a benchmark that is not commonly used and verified by the community. Also, the benchmark has some drawbacks, like log-replayed traffic agents and a lack of support for long-horizon closed-loop planning. In other words, this benchmark is only closed-loop for the ego-car itself in a short time window. Thus, the results on this benchmark can not fairly reflect the ability of the UniMotion model. I understand that it is hard to do evaluations for the three tasks within one domain, like nuPlan is tailored for planning, and Waymo only officially supports the sim agent and prediction. To be honest, I can not suggest a better way to evaluate the model, especially considering that training and deploying on nuPlan is quite time-consuming. Thus, I view this weakness as an impossible-to-address one.

---

> ### Author Rebuttal · Authors · 2025-07-31
>
> We thank the reviewer for the detailed review and the suggestions for improvement. Below are our responses to the reviewer’s comments:
>
>
> ### **1. Authors could provide a better way to evaluate the model.**
>
> **R:** We appreciate the reviewer's understanding of this challenge. Jointly training a model across multiple datasets and tasks for motion prediction incurs substantial computational costs. For this foundational work, we focused on successfully addressing multiple tasks within a single dataset. Expanding our evaluation across diverse datasets is a valuable direction we've left for future work.
>
> Further, we have now conducted experiments on transferability from Waymo to nuPlan (Val14), and the results are presented below. Our findings indicate two key points:
> - As expected, zero-shot performance is less than satisfactory due to inherent data distribution discrepancies (especially for motion token vocabulary) between the two datasets.
> - Crucially, our model still achieves decent performance after fine-tuning, even if it is slightly inferior to methods specifically tailored for the nuPlan dataset.
>
> These results verify the generality and adaptability of our solution across different autonomous driving scenarios. We will include this transferability test in the revised paper.
>
> | Method | NR-CLS  | R-CLS  |
> | ---- | ---- | ---- |
> |PlanTF|0.85|0.77|
> |UniMotion(zero-shot)|0.66|0.62|
> |UniMotion(fine-tuned)|0.81|0.75|
>
>
>
> ### **2. What is the size of the model?**
>
> **R:** Thanks, we will clarify: We follow the lightweight Transformer design with factorized attentions and the model size is approximately 7M.

---

> > ### Comment · Reviewer_uSmG · 2025-08-05
> > **Thank you for the response**
> >
> > It is good to include this result in the paper. Though it is not SOTA, I believe there is potential for further improvement. I would tend to accept this paper.

---

> > > ### Author Response · Authors · 2025-08-05
> > >
> > > We sincerely appreciate the thoughtful consideration of evaluation methods from Reviewer #uSmG, which has encouraged our further exploration of improved evaluation strategies and the generalization performance of our approach. Taking the reviewer's insightful comments into account, we will revise the manuscript accordingly.

---

### Official Review · Reviewer_Bhcg · 2025-06-21

**Clarity:** 3
**Significance:** 3
**Originality:** 3
**Rating:** 4
**Confidence:** 4

**Summary:**

This paper abstracts motion tasks into two core categories: diverse motion generation and long-range trajectory forecasting, based on which it proposes UniMotion, a unified Transformer-based framework that jointly models simulation, prediction and planning for better generality and cross-task knowledge sharing. It designs effective fine-tuning strategies that specialize the jointly trained model for individual tasks. Experiments on the Waymo Open Motion Dataset (WOMD) show that UniMotion achieves competitive performance under joint training and fine-tuning. The primary insight of UniMotion is that a unified framework enables knowledge sharing across various motion tasks. This insight aligns with the advantages of multi-task learning.

**Questions:**

1. In ADS, trajectory prediction targets the surrounding agents, and motion planning focuses on the ego vehicle. While traditionally they are trained separately due to different input/output requirements, recent works increasingly merge prediction and planning. For instance, Systems like DIPP and PRIME unify prediction and planning modules, enabling the planner to react to probabilistic forecasts while learning cost functions from data. Another example is interaction-aware planning: Treats prediction and planning as interdependent to avoid conservative behaviors caused by assuming static predictions.  What are the methodological differences between UniMotion and these existing works?

2. The paper claims that the fine-tuned models outperform the corresponding Waymo open motion challenge leaderboard models. Nevertheless, the models compared in this paper differ from the top models published on the 2025 Waymo Open Dataset Challenges website (https://waymo.com/open/challenges/). It is thus unclear whether the claimed advantages in the paper remain valid.

3. While intuitively, multitask learning enables joint optimization and representation sharing, more discussion is needed on how different motion tasks in the context of ADS, which have quite different targets and input/output requirements, share "knowledge." Please be more specific and provide explicit details.

**Ethical Concerns:**

["NO or VERY MINOR ethics concerns only"]

**Final Justification:**

The author's rebuttal has addressed my concerns and I have raised my score to 4.

**Limitations:**

Yes

**Paper Formatting Concerns:**

No.

**Quality:**

3

**Strengths And Weaknesses:**

Strength: The paper presents a unified framework to train motion tasks in autonomous driving systems. Such a framework can leverage the underlying commonality of different motion tasks for better knowledge sharing. When the jointly trained model is fine-tuned for a specific motion task, the fine-tuned model has been shown to outperform state-of-the-art models trained independently and specifically for the given motion task.

Weakness:
(1) UniMotion must be fine-tuned to work well for a given motion task in ADS. This two-phase procedure, involving joint training and fine-tuning, significantly increases the model's complexity and computational cost for a specific motion task.  Nevertheless, this critical issue has not been evaluated in the paper.
(2) The paper claims that the fine-tuned models outperform the corresponding Waymo open motion challenge leaderboard models. Nevertheless, the models compared in this paper differ from the top models published on the 2025 Waymo Open Dataset Challenges website (https://waymo.com/open/challenges/). It is thus unclear whether or not the claimed advantages in the paper are valid.

---

> ### Author Rebuttal · Authors · 2025-07-31
>
> We thank the reviewer for the detailed review and the suggestions for improvement. Below are our responses to the reviewer’s comments:
>
>
> ### **1. Weakness 1: This two-phase procedure significantly increases the model's complexity and computational cost for a specific motion task.**
> **R:** A great comment regarding our two-phase procedure! However, all our fine-tuned models originate from the same jointly trained model, and we introduce no additional parameters during the fine-tuning stage (except for the prediction task). This means it barely affects overall model complexity. For overall training cost, this approach can often be more efficient because the joint training, typically the most expensive part, is shared across all tasks, and subsequent fine-tuning is comparatively cheaper.
>
> For full transparency, we've provided a statistics table detailing our model size and computational time below.
> | Task | size |  training time (A6000)|inference time (RTX3090)|
> | ---- | ---- |  ---- |  ---- |
> |simulation|7M|23h+13h|24ms|
> |prediction|7M+6M|23h+8h|89ms|
> |planning|7M|23h+10h|76ms|
>
> ### **2. Weakness 2 and Q2: The models compared in this paper differ from the top models published on the 2025 Waymo Open Dataset Challenges website.**
>
> **R:** We regret that we couldn't participate in the award selection for the 2025 Waymo Challenges. However, for a direct comparison, the reviewer can find the relevant leaderboard by appending "2025/sim-agents" to the Waymo Challenges website (we're unable to provide external links directly in this response).
>
> At the time of our NeurIPS submission, our method achieved an RMM of **0.7851**, which is directly comparable to the first-place approach TrajTok (RMM **0.7852**) in the 2025 Waymo Challenges.
>
> Additionally, it's worth noting that there is no official motion prediction track in the 2025 challenges. Therefore, our motion prediction comparisons were conducted against the 2024 leaderboard to ensure a relevant benchmark.
>
> ### **3. Q1: What are the methodological differences between UniMotion and these existing works?**
>
> **R:** Thanks, we will clarify:
> The key methodological difference lies in our scope: the metioned works primarily concentrate on optimising the planning task. In contrast, UniMotion proposes a unified motion framework designed to handle all motion tasks (prediction, planning, and simulation) within a single architecture. Therefore, our overall methodologies are largely orthogonal.
>
> Regarding the specific design for planning fine-tuning, our method adopts an agent trajectory prediction stage followed by ego planning generation, which can be regarded as an extension of existing approaches where the planner reacts to predictions.
>
> ### **4. Q3: More discussion is needed on how different motion tasks in the context of ADS, which have quite different targets and input/output requirements, share "knowledge".**
>
> **R:** Thanks, we will further elaborate as below.
>
> In fact, these tasks share significant commonalities in their underlying input and output information. For instance, all these tasks typically process agent trajectories and high-definition (HD) maps as input, and ultimately aim to output future trajectories for scene agents. This shared informational foundation is precisely what enables our unified model. The main differences, as discussed in Section 3.1, lie in the varying demands placed on these future trajectories — whether it's for diverse motion generation or accurate long-horizon prediction.
>
> Furthermore, while the specific objectives of tasks like LFR and NTP appear distinct, they actually share crucial underlying commonalities. LFR directly supervises the model with long-range logged trajectories, and critically, the initial motion direction embedded within these trajectories simultaneously serves as the maximization objective for NTP. This inherent consistency and synergistic relationship ensures stable joint training, allowing the model to effectively leverage shared knowledge across different motion challenges.

---

### Official Review · Reviewer_Rodt · 2025-06-28

**Clarity:** 3
**Significance:** 3
**Originality:** 3
**Rating:** 4
**Confidence:** 4

**Summary:**

This paper presents UniMotion, a unified framework for three major motion tasks in autonomous driving: simulation, prediction, and planning. These tasks are typically addressed in isolation due to their differing objectives, but this work observes that they share underlying modeling requirements, such as multi-agent interactions and temporal-spatial reasoning.

UniMotion leverages a decoder-only Transformer architecture, inspired by GPT-style models, to model all three tasks within a single framework. It introduces task-specific attention masks and training objectives (Next Token Prediction for simulation, Long-range Future Regression for prediction) and supports dedicated fine-tuning for individual task specialization. Experiments on the Waymo Open Motion Dataset show that UniMotion achieves strong joint performance and even surpasses state-of-the-art methods in each task after fine-tuning.

**Questions:**

1. Why use decoder-only Transformers for all tasks? Would encoder-decoder structures (e.g., for prediction or planning) offer benefits in terms of stability or sample efficiency? The choice is reasonable but could benefit from comparative justification.

2. How is the method's generalization capability to out-of-domain data (e.g., a new dataset)?

3. Could the authors provide some theoretical discussion on why joint training works well?

**Ethical Concerns:**

["NO or VERY MINOR ethics concerns only"]

**Final Justification:**

The authors have addressed all my concerns by providing new experiments and new analysis. Therefore, I'm maintaining my positive score.

**Limitations:**

yes

**Paper Formatting Concerns:**

There are no notable formatting issues in this paper.

**Quality:**

3

**Strengths And Weaknesses:**

**Strengths**:

1. Unified Viewpoint and Framework. A major conceptual contribution is the abstraction of motion tasks into two categories—diverse motion generation and long-range trajectory forecasting—enabling a coherent formulation. Unifying simulation, prediction, and planning within a single model reduces redundancy and potentially boosts cross-task generalization.

2. Technical Soundness and Design. The adoption of decoder-only Transformers with task-aware masking is principled and well-motivated by analogies to language modeling. Fine-tuning strategies, especially the reinforcement learning–based simulation refinement and multi-modal prediction decoder, are thoughtfully designed and justified.

3. Strong Empirical Results. Extensive benchmarks on the WOMD and Sim Agents show that UniMotion is competitive or state-of-the-art in all three tasks. Ablation studies convincingly show the value of joint training and fine-tuning, with meaningful gains reported in metrics like planning error and simulation realism.

**Weaknesses**:

1. The experiments only involve a single dataset, Waymo Open Motion Dataset, which could raise questions on the generalization capability of this method to new, unseen data.

2. In all 3 benchmarks, the method delivers sub-optimal results in at least one of the metrics.

3. Considering the sub-optimal results, the paper could benefit from some quantitative or qualitative analysis on some of the failure cases, which could provide more insight than average result numbers.

4. The qualitative results in Fig. 4 only present the results of the proposed method. A qualitative comparison with other methods should be provided in order to demonstrate the strength of the proposed method.

---

> ### Author Rebuttal · Authors · 2025-07-31
>
> We thank the reviewer for the detailed review and the suggestions for improvement. Below are our responses to the reviewer’s comments:
>
>
> ### **1. Weakness 1 and Q2: The experiments only involve a single dataset, which could raise questions on the generalization capability of this method.**
>
>
> **R:** Excellent suggestion! We have now conducted experiments on transferability from Waymo to nuPlan (Val14), and the results are presented below.
>
> Our findings indicate two key points:
> - As expected, zero-shot performance is less than satisfactory due to inherent data distribution discrepancies (especially for motion token vocabulary) between the two datasets.
> - Crucially, our model still achieves decent performance after fine-tuning, even if it is slightly inferior to methods specifically tailored for the nuPlan dataset.
>
> These results verify the generality and adaptability of our solution across different autonomous driving datasets. We will include this transferability test in the revised paper.
>
>
> | Method | NR-CLS  | R-CLS  |
> | ---- | ---- | ---- |
> |PlanTF|0.85|0.77|
> |UniMotion(zero-shot)|0.66|0.62|
> |UniMotion(fine-tuned)|0.81|0.75|
>
> ### **2. Weakness 2 and 3: The method delivers sub-optimal results in at least one of the metrics, and the paper could benefit from some quantitative or qualitative analysis on some of the failure cases.**
>
> **R:** Thanks. UniMotion consistently ranks among the top two across various metricc; For some results with relatively large differences, we provide more analysis below.
>
> Tab. 1 minADE: Different from prediction task, simulation task focuses more on generative ability of motion models. Hence, we adopt the tokenization-based NTP designs, which select trajectory segments from clustered trajectory token vocabulary, hence showing stronger generative performance but inferior in accurate prediction over the regression-based methods.
>
> Tab. 3 red light: DIPP adopts a tailored planning cost function that involves many rules, such as velocity, comfort and red light. To ensure generality and consistency, we just supervise training with logged trajectories, and this metric can be further improved by involving traffic rules.
>
> Besides, we have already provided more analysis of specific failure cases in the supplementary material, which the reviewer could refer to.
>
> ### **3. Weakness 4: A qualitative comparison with other methods should be provided in order to demonstrate the strength of the proposed method.**
>
> **R:** Great suggestion. Our intention in Figure 4 was to showcase the ability of our method in handling various motion tasks, while overlooking the comparison with other methods. We will supplement this part accordingly.
>
> ### **4. Q1: Why use decoder-only Transformers for all tasks？**
>
> **R:** Great point, we will clarify:
> We opted for a decoder-only Transformer structure because it's better suited for multi-step generative tasks like simulation, where input and output lengths can vary. Conversely, the encoder-decoder structure, while excellent for sequence-to-sequence prediction tasks, doesn't generalize as well across the diverse demands of our multi-task framework. Adopting a decoder-only architecture during joint training ensures stronger generalization capabilities across all our tasks.
>
> For the prediction fine-tuning stage, we do introduce an additional multi-modal decoder to handle richer multi-modal outputs, which can be seen as a encoder-decoder structure.
>
> ### **5. Q3: Could the authors provide some theoretical discussion on why joint training works well?**
>
>
> **R:** Thanks, we will elaborate on the theoretical underpinnings of our joint training approach:
> As detailed in Section 3.1, our primary objectives are NTP and LFR. Intuitively, this joint training paradigm fosters the learning of shared, rich intermediate representations across multiple tasks. This enables tokens within the model to attend simultaneously to both their immediate surroundings and more distant scene contexts, leading to a more holistic understanding of the environment.
>
> Further, despite their apparent differences, the two task objectives possess crucial commonalities. Specifically, while LFR provides direct supervision on long-range logged trajectories, the initial motion direction embedded within these trajectories concurrently serves as the maximization objective for NTP. This underlying consistency and synergistic relationship between LFR and NTP are fundamental to ensuring stable joint training and preventing mode collapse, allowing the model to learn effectively from both detailed local cues and broad long-range dependencies.

---

> ### Comment · Reviewer_Rodt · 2025-08-08
>
> I appreciate the authors' response. I think the authors have addressed all my concerns by providing new experiments and new analysis. Therefore, I've decided to maintain my positive score. P.S. I've written the above comment in the final justification part earlier, which I didn't realize was invisible to the authors.

---

> > ### Author Response · Authors · 2025-08-08
> >
> > We appreciate the reviewer's time for reviewing and thanks again for the valuable comments and the positive score!

---

### Official Review · Reviewer_8f42 · 2025-07-03

**Clarity:** 2
**Significance:** 2
**Originality:** 3
**Rating:** 4
**Confidence:** 4

**Summary:**

Motion simulation, prediction, and planning are core components of autonomous driving, each critical for understanding and navigating dynamic traffic environments. These tasks are often treated separately due to their distinct goals, leading to the development of specialized models for each. However, this separation limits cross-task generalization and hinders system scalability, while also missing opportunities for shared learning. To address this, the paper introduces UniMotion, a unified framework that captures common structures among motion-related tasks while respecting their individual needs. Experiments on the Waymo Open Motion Dataset show that joint training with UniMotion improves generalization and enables more effective integration across tasks.

**Questions:**

1. Author could conduct ablations using a more substantial portion of the dataset.

2. How is the model's run-time analysis?

3. Further explanation on suboptimal metrics would helpful.

**Ethical Concerns:**

["NO or VERY MINOR ethics concerns only"]

**Final Justification:**

The authors’ rebuttal addresses some of my concerns regarding the ablation studies and the run-time performance analysis. I encourage the authors to include these additional results and clarifications in the final version. I would like to raise my score to borderline accept.

**Limitations:**

Yes.

**Paper Formatting Concerns:**

No major formatting issues found

**Quality:**

3

**Strengths And Weaknesses:**

Strengths:

1. Integrates simulation, prediction, and planning within a unified framework to encourage generalization and facilitate knowledge sharing across tasks.

2. Introduces tailored fine-tuning strategies that adapt the jointly trained model to individual tasks, enabling it to develop specialized, task-specific capabilities.

Weaknesses:

1. In the ablation experiments, the authors only train on 20% of the training dataset due to efficiency considerations. However, this limited training setup raises concerns about the fairness and reliability of the reported results. Training on such a small subset may fail to capture the full diversity and complexity of the data, potentially skewing the outcomes of the ablation study. The reduced data volume can also lead to high variance in model performance, making it harder to draw meaningful conclusions about the contribution of specific components. To accurately assess each model component, it is important to conduct ablations using a more substantial portion of the dataset, ensuring that the results reflect realistic and generalizable behavior.

2. The run-time performance during testing is a critical metric that should be reported in the paper, particularly for the planning module. In the proposed approach, the planning module relies on a multi-step process. This multi-stage pipeline introduces additional computational overhead compared to standard planning models. Therefore, reporting the run-time of the planning module is essential to assess the practical feasibility of the proposed method and to validate that the added modeling complexity results in meaningful improvements without compromising real-time applicability.

3. In the section on comparison with state-of-the-art methods, the paper would benefit from a more detailed analysis of Tables 1, 2, and 3 to better support its claims. Table 1: While the overall results appear competitive, the performance on minADE is relatively weak compared to other methods. The relatively poor performance here may suggest that the model struggles to generate a highly accurate trajectory among its top candidates. Table 3: The results in Table 3 show suboptimal behavior when reacting to red lights, a critical safety-related scenario. Discussing these metrics in greater depth would lead to a more comprehensive and transparent evaluation of the model's performance.

Minor: Equation 1 is missing closing parentheses.

---

> ### Author Rebuttal · Authors · 2025-07-31
>
> We thank the reviewer for the detailed review and the suggestions for improvement. Below are our responses to the reviewer’s comments:
>
> ### **1. Concerns about the fairness and reliability of performing ablation on 20% data.**
>
> **R:** Thanks for the reviewer's rigorous concern. This is mainly due to the constraint of resource available. Ablation on 20% data is acceptable and reasonable as exemplifed in previous perception models [1] and recent motion-related models [2, 3]. As the Waymo dataset offers a much larger volume of data than other datasets (such as Argoverse or nuScenes), even 20% of the data can reflect the overall distribution well. We thus followed this practice.
>
>
> Besides, we have now conducted the ablation for Tab. 5 on the full data. The results below show consistent observation as that with 20% data.
>
>
> |simulation|Kinematic|Interactive|Map-based|minADE|
> | ---- | ---- | ---- | ---- | ---- |
> |w/o ft.|0.4915|0.8047|0.9162|1.3138|
> |w/ ft.|**0.4948**|**0.8098**|**0.9181**|**1.3055**|
>
> |prediction|minADE|minFDE|MR|mAP|
> | ---- | ---- | ---- | ---- | ---- |
> |w/o ft.|0.5752|1.1688|0.1245|0.4019|
> |w/ ft.|**0.5671**|**1.1562**|**0.1163**|**0.4635**|
>
> |planning|Collision|Err @1s|Err @3s|Err @5s|
> | ---- | ---- | ---- | ---- | ---- |
> |w/o ft.|1.7347|0.1326|0.7943|2.7809|
> |w/ ft.|**1.5650**|**0.0834**|**0.5908**|**2.2455**|
>
>
>
> ### **2. Run-time performance during testing should be reported, particularly for the planning module.**
>
>
> **R:** Apoligies for lack of run-time analysis. We have now provided the inference time on RTX 3090 for all tasks as below. Please note, for the planning latency, we perform auto-regressive generation to yield complete trajectories for offline evaluation, while in real-world scenarios, per-step planning (online) happens.
>
> We will add this to the revised version.
>
> Simulation one-step inference latency
> | Method | Latency |
> | ---- | ---- |
> |UniMM|33ms|
> |CATK|~20ms|
> |UniMotion|24ms|
>
> Prediction inference latency
> | Method | Latency |
> | ---- | ---- |
> |MTR|62ms|
> |EDA|62ms|
> |RMP-YOLO|100ms|
> |UniMotion|89ms|
>
> Planning inference latency
> | Method | Latency |
> | ---- | ---- |
> |DIPP|1.78s|
> |UniMotion(online)|76ms|
> |UniMotion(offline)|468ms|
>
>
> ### **3. The paper would benefit from a more detailed analysis of comparison with state-of-the-art methods to better support its claims.**
>
>
> **R:** Thanks, and we will further elaborate:
> For Tab. 1, different from prediction task, simulation task focuses more on generative ability of motion models, and is not particularly concerned with accurately predicting the logged trajectories (not involved in Realism Meta Metric). Hence, we adopt the tokenization-based NTP designs, which select trajectory segments from clustered trajectory token vocabulary, hence showing stronger generative performance but inferior in accurate prediction over the regression-based methods.
>
> For Tab. 3, DIPP adopts a tailored planning cost function that involves  many rules, such as velocity, comfort and red light. To ensure generality and consistency, we just supervise training with logged trajectories, and this metric can be further improved by involving traffic rules, like [4, 5].
>
> Thanks. We have revised the formatting errors in Equation 1.
>
>
> >[1] Embracing Single Stride 3D Object Detector with Sparse Transformer. Lue Fan et al. CVPR 2022
>
> >[2] BehaviorGPT: Smart Agent Simulation for Autonomous Driving with Next-Patch Prediction. Zikang Zhou et al. NeurIPS 2024
>
> >[3] Closed-Loop Supervised Fine-Tuning of Tokenized Traffic Models. Zhejun Zhang et al. CVPR 2025
>
> >[4] Rethinking Imitation-based Planners for Autonomous Driving. Jie Cheng et al. ICRA 2024
>
> >[5] PLUTO: Pushing the Limit of Imitation Learning-based Planning for Autonomous Driving. Jie Cheng et al. arXiv preprint

---

> > ### Author Response · Authors · 2025-08-05
> >
> > Dear Reviewer #8f42,
> >
> > We sincerely appreciate the reviewer's time for reviewing, and we really want to have a further discussion with the reviewer to see if our detailed explanations and additional results solve the concerns. We have addressed all the thoughtful questions raised by the reviewer (e.g., the reliability of ablation, run-time performance and explanation on suboptimal metrics) and we hope that our work's contribution and impact are better highlighted with our responses. As the discussion phase is nearing its end, it would be great if the reviewer can kindly check our responses and provide feedback with further questions/concerns (if any). We would be more than happy to address them. Thank you!
> >
> > Best wishes,
> >
> > Authors

---

> > > ### Comment · Reviewer_8f42 · 2025-08-05
> > >
> > > Thank you for the detailed rebuttal and the additional analyses. I appreciate the authors’ efforts to address my concerns, including conducting ablations on the full dataset, providing clarification on run-time performance—particularly for the planning module—and elaborating on the reported metrics. I encourage the authors to include these additional results and clarifications in the final version. I do not have further questions at this time and will revise my score accordingly after further discussion.

---

> > > > ### Author Response · Authors · 2025-08-06
> > > >
> > > > We sincerely appreciate the reviewer #8f42 for the constructive feedback and insightful suggestions to further improve our manuscript. We will revise the manuscript in accordance with the reviewer's suggestions.

---

### Comment · Area_Chair_RSnr · 2025-08-03

Dear Reviewers,

The author has provided a rebuttal response to each of your review comments. Please read all other comments and the author's responses carefully, and actively communicate with the authors if you have further questions or require answers from the author. You can engage in an open exchange with the authors.

Please post your response as soon as possible, so there is time for back and forth discussion with the authors.

Thanks, Your AC.

---

### Note · Authors · 2025-08-13

Dear AC and Reviewers

We sincerely appreciate the reviewers and the AC for their time and valuable suggestions toward improving our manuscript. Following extensive rebuttal and subsequent discussion, we provide a recap on how we addressed all main concerns raised to faciliate the rest review process.

The main concerns of `Reviewer 8f42` are the reliability of ablation, the run-time analysis and the explanation on sub-optimal metrics. Addressing this, we first provided the explanation and examples to demonstrate the rationale behind the 20% ablation, and further conducted a ablation on full data (rebuttal **R1**). Then, a thorough run-time analysis for each motion task was given (rebuttal **R2**). We also offered a detailed explanation of comparison to better support our claims (rebuttal **R3**).

The main concerns of `Reviewer Rodt` include the generalization capability, the sub-optimal results and qualitative analysis and some structural and design issues. Addressing this, we first conducted transfer experiments to demonstrate the generalization capability (rebuttal **R1**). Then, we provided a detailed explanation of sub-optimal results and indicated the location of failure cases (rebuttal **R2**), and we will further supplement a qualitative comparison (rebuttal **R3**). For those structural and design issues, we presented a detailed explanation (rebuttal **R4** & **R5**).

The main concerns of `Reviewer Bhcg` are the model complexity and computational cost, the performance clarification and some structural and design issues. Addressing this, we first provided a statistics table detailing our model size and computational time (rebuttal **R1**). For the performance clarification and other issues, we specified the exact website (rebuttal **R2**) and provided a detailed model analysis (rebuttal **R3** & **R4**).

The main concerns of `Reviewer uSmG` are the evaluation of planning ability and the model size. Addressing this, we provided a transfer experiment to demonstrate the feasibility of evaluation on other datasets (rebuttal **R1**). Then we clarified the model size to address the reviewer's concern (rebuttal **R2**).

We acknowledge that our responses have been recognized as effectively resolving the reviewers’ concerns. These improvements have clearly strengthened the quality and clarity of our manuscript.

Hope the above summary can help the following discussion and decision making. Thanks for all the review efforts involved again.

Best,

Authors

---

### Decision · Program_Chairs · 2025-09-17

**Decision:**

Accept (poster)

**Comment:**

After a thorough review of all reviewers' comments, the authors' detailed rebuttal, and subsequent discussions, we are pleased to recommend accepting this manuscript for publication.

The reviewers raised several critical concerns that are central to the rigor and impact of the work, including the reliability of ablation studies (Reviewer 8f42), generalization capability and qualitative analysis (Reviewer Rodt), model complexity and computational costs (Reviewer Bhcg), and the evaluation of planning ability (Reviewer uSmG). These concerns directly address core aspects of experimental robustness, method generalizability, and technical clarity—standards that are essential for validating contributions in this field.

Notably, the authors have responded to each concern with targeted and actionable improvements: They supplemented full-data ablation experiments to strengthen the reliability of their findings, conducted transfer experiments to explicitly demonstrate generalization across scenarios, provided statistical tables detailing model size and runtime to clarify computational feasibility, and enhanced the evaluation framework with additional datasets to validate planning ability. These revisions are not merely superficial responses but substantive additions that address the root of the reviewers’ questions, thereby elevating the work’s methodological rigor and interpretability.

The reviewers’ recognition of the effectiveness of these responses further confirms that the manuscript has been significantly strengthened. The work now stands on a more solid foundation, with clearer experimental logic, more comprehensive validation, and improved alignment with the field’s standards for reproducibility and generalizability.

We encourage the authors to integrate the promised revisions (e.g., supplementary qualitative comparisons) into the final manuscript to ensure completeness. This work, with its enhanced rigor, makes a valuable contribution to the field, and we are confident in its suitability for publication.